# From challenge to growth: A qualitative study of parental adaptation to Autism Spectrum Disorder

Chiara Fante[1]*, Fabio Fontana[1,2], Francesca Capelli[3], Barbara Dioni[3], Mattia Pezzi[2,4], Cinzia Raffin[3], Raffaele De Luca Picione[5], Alessandro Musetti[2]*

1 Institute for Educational Technology, National Research Council, Via de Marini 6, Genoa, Italy, 2 Department of Humanities, Social Sciences and Cultural Industries, University of Parma, Borgo Carissimi 10, Parma, Italy, 3 Fondazione Bambini e Autismo Onlus, Via Vespucci n.8/a, Pordenone, Italy, 4 Department of Biological and Health Psychology, Faculty of Psychology, Autonomous University of Madrid, Calle Ivan Pavlov 6, Madrid, Spain, 5 Faculty of Law, Giustino Fortunato University, Viale Raffaele Delcogliano 12, Benevento, Italy

* chiara.fante@cnr.it (CF); alessandro.musetti@unipr.it (AM)

## Abstract

Raising a child with Autism Spectrum Disorder (ASD) profoundly affects family dynamics and parental well-being. While research has often focused on stress and negative outcomes, less is known about the processes and resources that foster adaptation and growth in parents. This qualitative study explored the adaptation process among parents of children with ASD, examining both challenges and transformative experiences, and identifying key personal and contextual factors that support or hinder adjustment. Thirty-six parents (19 mothers, 17 fathers) of children aged 5–11 years with ASD (severity level 2 or 3, DSM-5) were recruited from two health care centres for ASD in Northern Italy. Semi-structured interviews were conducted using a guide developed according to an established methodological framework for qualitative interviews and analysed following Braun and Clarke's framework. Themes were organised into three conceptual domains (i.e., Outcomes, Resources, and Challenges) reflecting the main research aims. Parents reported both emotional distress and personal growth, including improvements in family functioning and child development (Outcomes). Facilitating factors included social support, access to information, professional interventions, and parental self-efficacy (Resources). Barriers included difficulties with services, family conflicts, social stigma, and maladaptive coping strategies (Challenges). Parental adaptation to ASD emerged as a dynamic, ongoing process of negotiation between challenges and resources, often leading to transformative experiences and personal growth. The findings support the implementation of systematic intervention strategies aimed not only at reducing parental stress, but also at empowering parents and promoting the development of adaptive resources.

**Data availability statement:** The minimal dataset required to replicate the study's findings—including the complete codebook and all de identified excerpts that directly support the analytic claims—has been made available in the Supporting Information files, in accordance with PLOS ONE requirements for qualitative human participant data. The full raw interview transcripts cannot be publicly shared because they contain potentially identifying and sensitive personal information from parents of children with autism. Public release would not be compatible with the conditions of informed consent and the study's approved ethical protocol. In line with PLOS ONE guidelines for qualitative research involving human participants, additional fully de identified data excerpts may be shared upon reasonable request for the purpose of scientific verification. To comply with PLOS ONE policy—which does not allow any author to serve as the sole access point for restricted data—requests for access to additional de identified materials should be directed to an independent institutional contact: University of Parma – Data Protection Officer (DPO) Email: dpo@unipr.it Website: https://www.unipr.it/ The DPO will receive external data access requests and forward them to the research team, ensuring an independent and stable institutional point of contact. Qualified researchers who meet ethical and confidentiality requirements may obtain access to additional de identified data in accordance with participant consent, privacy safeguards, and institutional regulations.

**Funding:** The author(s) received no specific funding for this work.

**Competing interests:** The authors have declared that no competing interests exist.

## Introduction

Raising a child with Autism Spectrum Disorder (ASD) has a significant impact on family dynamics [1] and parenting [2], potentially affecting the mental and physical health of parents [3] and siblings [4,5]. The period surrounding diagnosis is often marked by acute emotional upheaval, as parents experience shock, and psychological distress [6–8]. This critical juncture necessitates a reorganization of daily routines and family priorities, prompting parents to seek new coping strategies and support systems. Following diagnosis, parents may experience a loss of expectations and uncertainty about the future [9,10], embarking on a complex adaptation process shaped by both personal and external factors. Recent literature emphasizes that interventions should consider parental outcomes alongside those of children, both as indicators of intervention success and as explicit goals [7]. Understanding the parental adaptation process and the factors influencing it is therefore crucial for enhancing the effectiveness of interventions.

### ASD and parental adaptation process: Beyond stress and negative outcomes

Parental adaptation following an ASD diagnosis is a dynamic and multifaced process, reflecting ongoing efforts to maintain family balance through the reorganisation of daily life, roles and responsibilities, as well as the mobilisation of emotional and environmental resources [11]. Many studies have focused on the challenges faced by parents and the negative outcomes associated with raising a child with ASD. Cappe and colleagues [12] emphasize that parental quality of life (QoL) is crucial for understanding psychological adjustment and assessing adaptation to a child's disability. Vasilopoulou and Nisbet's [13] review found that parents of children with ASD generally report lower QoL compared to those with typically developing children. Several factors have been associated with lower parental QoL, including personal factors (e.g., maladaptive parental coping strategies), child characteristics (e.g., ASD severity), and environmental factors (e.g., low availability of social and professional support).

Alongside studies focusing on critical outcomes, research has increasingly explored positive adaptation in parents, drawing on a broader body of evidence that suggests stressful or potentially traumatic events can promote personal growth [14]. This concept, defined as a positive psychological transformation arising from overcoming difficult circumstances and life events, has been described using different conceptual labels that reflect partially different but interconnected theoretical perspectives, including *post-traumatic growth* [15,16], *stress-related growth* [17], and *post-crisis growth* [18,19]. Several studies have documented this process of growth among parents facing complex challenges, such as caring for children with chronic illnesses or disabilities.

In the context of ASD, many parents report that raising children with ASD provides new perspectives on life, reshapes their values and ethics, and foster personal development [19]. Some families even describe their child's condition as a source of fulfilment, happiness, intimacy, and strength [20], finding positive meaning in their caregiving role [8,11,21].

A broad array of personal and contextual factors can shape and support positive outcomes. Among relational and environmental variables, a supportive context plays a crucial role in reducing stress and fostering adaptive responses to challenges [1,11]. For parents of children with ASD, social support has been linked to lower maternal stress, reduced levels of depression and anxiety, and an overall better QoL [3,11,22]. Interventions targeting children can also be an essential resource in shaping the parental adaptation process. For instance, parental involvement in these interventions has been shown to foster the acquisition of adaptive strategies and to enhance parents' confidence and sense of self-efficacy [23,24]. Among personal resources, parental self-efficacy, defined as parents' belief in their ability to effectively raise their child [25], is as a key factor that positively influences parenting outcomes [11]. Higher levels of self-efficacy not only increase parents' confidence in their role but also help to reduce feelings of guilt [3]. In addition, coping strategies are pivotal in the adaptation process. Problem-focused coping, particularly cognitive restructuring, is associated with lower parental stress and more positive outcomes [3,7,11].

To date, however, few studies have examined the specific mechanisms that foster personal growth in parents of children with ASD. Emerging evidence suggests that factors such as general self-efficacy, adaptive cognitive emotion regulation strategies [14], and active attribution of meaning to adversity [18] may play a key role in this process. This gap highlights a broader limitation in the literature on parental adaptation to ASD. While recent research has begun to address positive outcomes, such as improvements in family QoL [8,11], the majority of studies continue to focus primarily on negative outcomes [1,11]. Moreover, many investigations do not distinguish ASD from other disabilities, nor do they systematically examine positive and negative outcomes as distinct yet coexisting phenomena within families [11].

Furthermore, there is limited attention to the dynamic processes and strategies that parents adopt over time, as well as the contextual and personal factors that facilitate or hinder adaptation. Gaining deeper understanding of these processes is essential for elucidating how parents cope with ongoing challenges and gradually adapt to raising a child with ASD. Such insights can move the field beyond outcome-based measures, offering a more nuanced perspective on the lived experiences and resilience of these families.

Given these limitations, the present study aims to advance understanding of parental adaptation following an ASD diagnosis by qualitatively exploring the lived experiences of both mothers and fathers. Specifically, we focus on how parents navigate the interplay of challenges and resources throughout their adaptation journey, and how these factors shape both immediate and long-term outcomes for families. By examining not only the difficulties but also the personal and contextual resources that support growth and resilience, this study seeks to move beyond outcome-based measures and provide a nuanced perspective on the dynamic processes underlying parental adaptation. Ultimately, our goal is to identify key mechanisms and strategies that can inform more effective, family-centred interventions and support systems for parents raising a child with ASD.

## Materials and methods

### Participants

Participants were recruited from two health care centres in Northern Italy that provide comprehensive, long-term support to individuals with ASD. The inclusion criteria were as follows: a) being the parent of a child with a formal diagnosis of ASD for at least two years prior to data collection; b) having a child aged between 5 and 11 years old, with a diagnosis classified as severity level 2 or 3 according to the Diagnostic and Statistical Manual of Mental Disorders–Fifth Edition–Text Revision [26]. Specifically, Level 2 indicates a significant need for support due to marked difficulties in social communication and the presence of restricted and repetitive behaviours, while Level 3 refers to more profound impairments requiring very substantial support and significantly interfering with daily functioning. The selected age range ensured a relatively homogeneous developmental period, avoiding challenges specific to adolescence or adulthood, which may reflect different parenting experiences. The exclusion of Level 1 ASD diagnoses aligned with the aim of focusing on parents of children

with substantial support needs, whose experiences may differ considerably from those of parents of children requiring lower levels of assistance.

Additional criteria included: c) the child was receiving multidisciplinary treatment for ASD, involving collaboration among professionals; d) no other family members had disabilities or chronic conditions; and e) proficiency in Italian language to understand and participate in the interview.

Compliance with the inclusion criteria was verified through preliminary screening questions before the interview. All participants provided written informed consent prior to their inclusion in the study. Data collection was conducted from 2 March 2022–16 July 2022.

A total of 42 eligible parents were contacted and informed about the objectives of the study. Four potential participants (three mothers and one father) declined due to lack of time, and two parents (one mother and one father) initially agreed to participate but later withdrew because of bereavement in the family.

The final group of participants consisted of 36 parents ($n = 19$ mothers, 52.8% and $n = 17$ fathers, 47.2%), aged between 29 and 51 years old, of 22 children with ASD. The age of the children ranged between 5 and 11 years. Among the 22 families recruited, 7 had one child, 14 had two children, and 1 had three children. All participants had been living in Italy for at least 10 years: mothers, 15 were born in Italy, two in Eastern Europe, and two in South America; among the fathers, 14 were born in Italy, one in Eastern Europe, one Western Europe, and one in South America. Additional demographic information is reported in Table 1. Of the children with ASD, two had intellectual impairment, nine had a language impairment, and ten children had both intellectual and language impairments. None of the siblings had a clinical diagnosis.

## Procedure

The data for this study were drawn from a broader investigation examining factors influencing QoL of parents of children and adolescents with ASD [27]. While the primary objective of the larger project was to identify which aspects of interventions directed at children may affect parental QoL, the project was intentionally designed to obtain thorough narratives that describe participants' experiences in detail. A semi-structured interview guide was developed to explore a range of relevant topics, as detailed below. Ethical approval for the study was obtained from the Research Ethics Board of the University of Parma (prot. 0064713).

## Development of interview guide

A semi-structured interview guide was developed, consisting of a series of questions designed to elicit authentic and personal responses from participants about their experiences with ASD. The guide was also designed to cover different areas of inquiry, in line with the overall aims of the study. Following the methodological framework proposed by Kallio et al. [28], the guide was structured on two levels: general questions, followed by prompts designed to elicit further discussion. The interview covered four main areas: 1) History with ASD (i.e., the parental journey from diagnosis, with particular attention to the impact on personal life); 2) Communication (i.e., managing ASD within the family and in interactions with the external environment); 3) Interventions (i.e., parents' experiences with support services and their perceptions of their effect on the family system); 4) Personal reflection (i.e., daily experiences of living with and managing ASD).

A pilot phase was conducted with two parents (one mother and one father) to assess the clarity, relevance and completeness of the topics covered in the guide. Based on their feedback, only minor adjustments were made before finalizing the instrument.

Data collection was carried out by two clinical psychologists. Interviews were scheduled according to parents' availability, and each participant was informed of the purpose of the study and encouraged to respond as openly as possible. Interviews lasted between 40 minutes and approximately two hours. Each interview was recorded and transcribed verbatim.

**Table 1. Socio-demographic features of the sample.**

| Caregivers (*N* = 36) | | |
|---|---|---|
| Mothers: Fathers | | 19:17 |
| Caregiver age in years *M* (*SD*) | | 42.69 (4.91) |
| Education *n* (%)<br>Na = 1 | Middle school diploma | 2 (5.7%) |
| | Secondary school diploma | 18 (51.4%) |
| | Bachelor's/ master's degree | 15 (42.9%) |
| Occupational status *n* (%)<br>Na = 1 | Employee | 23 (65.7%) |
| | Housewife | 5 (14.3%) |
| | Self-employed | 4 (11.4%) |
| | Student | 1 (2.9%) |
| | Unemployed | 2 (5.7%) |
| Family status *n* (%) | Married/cohabiting parents | 33 (91.7%) |
| | Separated/divorced parents | 3 (8.3%) |
| **Children (*n* = 22)** | | |
| Child with ASD gender (male: female) | | 17:5 |
| Child with ASD age in years *M* (*SD*) | | 8.95 (1.65) |
| Birth order of the child with ASD (%) | Only child<br>Eldest child<br>Youngest child | 7 (31.8)<br>9 (40.9)<br>6 (27.3) |
| Households with one child (%) | | 7 (31.82) |
| Households with two or more children (%) | | 15 (68.18) |
| Gender of the first sibling without ASD (male: female) | | 6:9 |
| Gender of the second sibling without ASD (male: female) | | 1:0 |
| **Sibling age in years M (SD)** | | **7.88 (5.01)** |

## Data analysis

This study adopted a codebook approach to Thematic Analysis (TA), situated within the broader family of TA methods that share similar procedures but differ in epistemological assumptions and research values. The codebook approach was selected for pragmatic reasons, as it supported collaborative teamwork, provided a structured yet flexible framework for organizing the material, and facilitated systematic mapping of the dataset to meet the study's descriptive aims [29]. Unlike approaches based on coding reliability, the codebook did not function as a tool for assessing coding reliability; rather, it served as an iterative guide that evolved throughout the analytic process [29; 30].

The analysis was conducted in line with the 'Big Q' qualitative framework proposed by Braun & Clarke [29; 30], which views knowledge production as inherently interpretative. Within this perspective, researcher subjectivity is considered a resource, not a bias to suppress. Accordingly, no formal reliability procedures (such as inter-coder agreement statistics or consensual coding) were used, as these would not align with Big-Q assumptions. Instead, the research team engaged in reflective dialogue, ongoing discussion, and critical interpretation to co-construct a rich and multifaceted understanding of the data.

During the early familiarisation phase, researchers read the interview transcripts multiple times. Four interviews were selected as particularly suitable for preliminary coding, given their narrative richness (two with mothers and two with fathers). Each member of the research team independently coded these transcripts, generating preliminary conceptual labels (codes) that captured key elements of meaning. Through reflexive dialogue and peer debriefing, these initial labels were developed into a preliminary codebook.

The full dataset was then coded by two authors (C.F and F.F.) using this evolving codebook, which was iteratively refined to chart and map the developing analysis [29]. Coding decisions were regularly discussed and reviewed in team meetings and peer debriefing sessions, adopting a collaborative and reflexive approach to data interpretation [30].

At the conclusion of the coding process, similar or related codes were grouped into preliminary sub-themes and themes, which were developed inductively from the coded data and then collaboratively reviewed and refined. Specifically, the themes were actively generated and constructed by the research team, reflecting the researchers' active role in the analytic process. In line with the descriptive focus of the study, these themes were conceptualized as "topic summaries", structured overviews that unite participants accounts around specific conceptual domains. These summaries were organized into three broad domains- Outcomes, Resources, and Challenges- to provide a clear and transparent narrative of the adaptation process. The final coding system comprised 67 codes, grouped into 7 main themes. Data analysis was conducted using MAXQDA, a software application for qualitative data analysis [31]. The complete system of codes, including definitions and organisation into sub-themes and themes, is available in the supplementary materials (S1 Table).

## Results

The themes and sub-themes identified through the analysis, and organized within the three overarching conceptual domains, are visually presented in Fig 1 and described in detail below.

### Outcomes of the adaptation process

Three main themes were identified: 1) *emotional distress*, which includes codes relating to the emotional and psychological difficulties that parents continue to face in adapting to their child's condition; 2) *personal growth and transformation*, which includes codes referring to positive psychological, emotional or relational changes in parents after dealing with their children's ASD; and 3) *family and child improvements*, which collects codes relating to improvements reported by parents in a range of domains.

**Theme 1: Emotional distress.** Some parents expressed emotional distress during the interview in relation to their condition or psychological and emotional difficulties in accepting their children' diagnosis of ASD. In some cases, they reported a sense of resignation when faced with the demands of managing their child's needs. They expressed feelings of frustration, inevitability, often accompanied by an attitude of defeat.

A father said: "*We don't even think about the future that much, because there's not much we can do. When you receive news like that* [the diagnosis]*, the only thing you're left with is the certainty that you're going to have an unhappy life*".

Some parents viewed the challenges as an ongoing aspect of their lives and did not anticipate significant improvement or change. Concerning the future, some participants conveyed uncertainty and concern, noting ongoing questions about their children's capacity for independent living and future care arrangements.

A mother noted: "*After that age, at six years old, it's not normal for a child not to talk or do the things that other children do. So, we would say that it's quite a serious concern, a source of anxiety, especially for the future, not so much for the present, but for the future*".

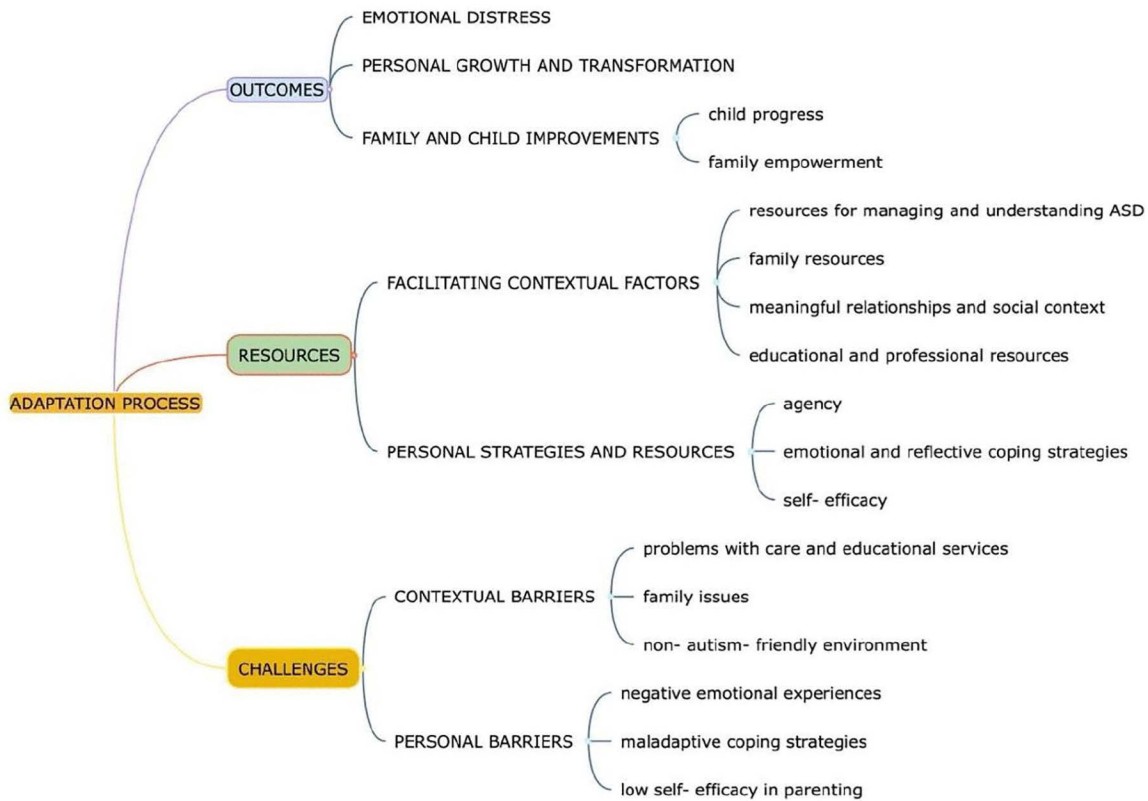

**Fig 1. Themes and sub-themes identified in the analysis, organized within three overarching conceptual domains.**

Some participants reported a sense of difference from other parents, often related to challenges with managing social situations and interactions.

A mother said: "*So you live with autism, but this is actually your 'normal' life. Since I don't have other children to tell me what it's like... I can imagine it, because I see it from the outside, but it's not the same as living it. It's like a parent who has a normal child and can't understand what I'm going through*".

**Theme 2: Personal growth and transformation.** Parents commonly described a significant change from initial shock, anxiety, and disorientation to greater emotional stability and empowerment. Acceptance of the diagnosis was pivotal, enabling a shift from crisis management to a more constructive and future-oriented approach.

A father noted: "*At first it was tough, but then, little by little, as we went along... now I'm really happy with my son. It's a whole new world... yes, it has opened up a completely different perspective for me*".

Furthermore, facing ASD-related challenges can lead to growth in personal values, increased appreciation for small successes, and strengthened their commitment to supporting others in similar situations. While acknowledging the emotional difficulty, parents reported a shift in perspective characterised by greater compassion and sensitivity.

A mother stated: "*So, you also become more sensitive... as a family, as a person, towards people who are going through a difficult time. And then maybe you also meet people you never expected to meet*".

Some parents also expressed a sense of hope about their children's future, noting progress beyond expectations and believing greater independence and social inclusion are achievable goals.

A mother stated: "*When you see these changes, you say - No, wait... things can change- If they change every day, as I said, you don't set limits because every day you can add something, like in a puzzle*".

Recognition of their child's uniqueness and unconditional acceptance were also highlighted, as well as the development of more authentic and intimate relationships.

A mother said: "*I look at my son and sometimes I think to myself... how wonderful that it is like this*".

Finally, some parents reported that their relationships became more authentic and intimate as a result of their experience with their children' ASD.

A mother affirmed: "*This life with S. has led us to meet wonderful people whom I would never have had the opportunity to meet otherwise, so... in short, it has had a very, very strong impact*".

**Theme 3: Family and child improvements.** This theme consists of two sub-themes: *child progress* and *family empowerment*.
*Child progress.* In general, parents reported substantial improvements in their children's overall functioning following interventions. They referred that children become more capable, independent, and integrated into society. Parents also noted greater confidence in their child's growth and development, often relating these changes to the interventions received.

A father said: "*When you see improvements, you're motivated to do even more, right? We do therapy and I see that he didn't greet me before, but now he does, so, how can I put it... you feel motivated, you have new energy*".

Many parents observed significant behavioural improvements in their children over time, indicating that appropriate intervention can enhance manageability. Changes in communication are also documented, including developments in language skills and expressive abilities, as well instances of transition from non-verbal to verbal communication. Improvements in communication supported greater understanding and interaction between parents and children. Additionally, participants noted increases in children's autonomy and independence, with some becoming more self-sufficient in daily activities.

A mother affirmed: "*Well, first of all, the fact that he started talking, so I can understand what he needs, which I think is essential. You tell me what you want, how you feel at that moment... Actually, I always try to encourage this kind of communication at home too. Then, of course, his autonomy has also increased*".

Over time, parents reported changes in their children's social skills, which may affect social relationships and participation in leisure activities. As children became more able to express their needs and participate in group settings, families experienced differences in social interactions.
*Family empowerment.* Many parents reported an overall improvement in their QoL following the ASD diagnosis: while the initial period after diagnosis was frequently challenging, as children gained autonomy and communication skills, family

routines became more manageable and overall well-being improved. Additionally, parents frequently noted advancements in their autism literacy; increased familiarity with ASD tended to lead to greater awareness and improved understanding of their children.

A mother stated: "*When you start studying and learning more about autism, everything starts to have a name... This is fundamental, and I always emphasise this to other parents who are starting this journey, because the more you know, the more you understand your children's world and the more you understand that it is not... a sentence*".

In addition, the greater independence of children allowed parents to have time for themselves outside the family and to better manage work and family commitments.

A mother said: "*In the morning, unlike before, I get up, have breakfast sitting down, drink a coffee... Before, most of the time, I couldn't drink coffee because it wasn't ready... My quality of life is improving day by day, increasingly*".

Parents also reported feeling less emotional stress associated with ASD in various situations, particularly considering the significant improvements in their children's functioning and the targeted interventions.

## Resources in the adaptation process

Two main themes were identified under this domain: 1) *facilitating contextual factors*, and 2) *personal strategies and resources supporting adaptation*.

**Theme 1: Facilitating contextual factors.** This theme consists of four sub-themes: *resources for managing and understanding ASD*; *family resources*; *meaningful relationships and social context*; and *educational and professional resources*.

*Resources for managing and understanding ASD.* Access to quality information has been crucial for parents adapting to children with ASD: understanding ASD traits helped parents handle challenges, while augmentative and alternative communication (AAC) tools supported communication, expression of needs, and reduced anxiety and behavioural issues. Parents reported that AAC tools strengthened relationships and improved quality of life, despite requiring initial effort to implement.

One father said: "*Augmentative communication with images has had a huge impact. We noticed that S. is very visual, so showing him activities through images or written words has been a big help*".

*Family resources.* This theme encompasses codes pertaining intra-family factors that facilitate the management of daily life and parental adaptation. Parents referred the importance of family support, frequently referencing their family of origin as a key source of assistance. In addition, parents identified siblings as valuable contributors; siblings' presence is also seen as promoting social integration and acceptance of the family within broader community contexts. In addition, some parents emphasized the significance of strong partner relationships and mutual understanding within couples for effectively navigating complex circumstances. Some fathers specifically acknowledged mothers' active participation and problem-solving skills as strengths. In such instances, that mothers were described as proactive in pursuing early diagnosis and interventions, as well as engaging in information-seeking activities through reading, joining support groups, and building networks with families in similar situations.

A father said: "*I am grateful to my wife, she did everything else at home... she worked hard every single day with activities, homework, she looked after him, she still does, I mean, she did so many things*".

Children with ASD also served as resources, as many parents identified positive characteristics such as adaptability, curiosity about their environment, and sociability.

*Meaningful relationships and social context.* Social relationships have been vital for parents of children with ASD. Both mothers and fathers reported that friends and community members provided practical and emotional support in coping with daily challenges and emphasized the need to maintain relationships in the social context. Specifically, informal support among parents of children with ASD played a key role: connecting with others who have had similar experiences is considered essential for coping. Such interactions reduced feelings of isolation, provided emotional validation, and gave access to practical advice and coping strategies. Informal settings, such as waiting rooms or chance encounters, have been described as particularly conducive to these exchanges. Although professional assistance has been recognized as essential, many parents emphasized that talking to other people who have been through the same experience offered a unique and valuable perspective, particularly during the early stages of accepting the diagnosis.

> A father noted: "*There is this sharing between parents... I used to talk to that father or mother over a coffee, in a moment of confidence. In my opinion, sharing experiences between parents in this way... I think it can help beyond parent training*".

An additional factor that contributed to family adjustment was the social inclusion of children. In several interviews, parents reported instances where their children with ASD are accepted by their peers. While these experiences varied in consistency, they were viewed as significant and may support both children's sense of belonging and well-being, as well as influence parents' perceptions of normalcy.

*Educational and professional resources.* Targeted interventions within educational and healthcare settings provided valuable support not only to children with ASD but also to their families. Many parents emphasized the role of intervention programs and access to professional expertise in promoting family well-being.

> A mother said: "*Relying on experts is a great help, especially in difficult times, when you feel unable to deal with a critical situation involving your child: knowing the strategies suggested by professionals and knowing that you are able to manage the situation...*".

Parent training and active involvement in intervention were widely recognized as essential components in the effective support of families. Key benefits included increased knowledge of ASD, acquisition of practical approaches, and improvements on overall family dynamics. Additionally, participants underscored the role played by professionals in improving their lives. Furthermore, the standard of the educational environment was recognized as a critical factor influencing the sense of support and security experienced by families.

**Theme 2: Personal strategies and resources for adaptation.** Overall, parents described a range of strategies and resources for managing ASD that have supported their adaptation process. Some of the codes identified were clustered under the sub-theme *agency*; other codes were organized under the sub-theme *emotional and reflective coping strategies*. The code *self-efficacy* was considered an independent sub-theme.

*Agency.* This sub-theme includes codes that refer to parents' ability to interact intentionally and proactively with their children. It therefore includes diverse, targeted and intentional actions that parents used to address daily challenges.

In many cases, parents reported using active problem-solving strategies, taking deliberate efforts to identify practical solutions that include planning, decision-making, and taking concrete steps to manage or resolve difficulties.

> A father stated: "*If you pretend that nothing happened... you can close your eyes, you can pretend to live the same life as before, go to the pub with your friends... but the problem will remain there for the next few years of your life. So, you have to accept it, roll up your sleeves and invest a lot to improve the child's situation, one step at a time*".

Furthermore, many participants described an ongoing process of experimentation and adaptation when seeking effective strategies to support their children's development, adopting a flexible and patient method. This approach is also noted in social contexts; gradual exposure and initially avoiding highly stimulating environments promoted better adaptation and reduced discomfort for children. Parents adjusted social activities by selecting quieter or more structured settings and aiming to progressively encourage more socially appropriate behaviours before broader social integration.

Parents reported using various strategies when seeking interventions for their children. Rather than only following professional recommendations, they searched for effective interventions, specialized professionals, and therapies, persisting despite initial challenges within the healthcare system. Parents also noted their involvement in coordinating among different specialists and services, such as psychologists, therapists, and educators, to establish an integrated support network. Additionally, they described efforts to obtain information about ASD, often prior to formal diagnosis, through independent research and consultations.

*Emotional and reflective coping strategies.* Some participants emphasized the crucial role of emotional support in coping with an ASD diagnosis. In these cases, parents reported actively seeking support to manage their emotions within their family or wider social context. Other strategies aimed at effectively managing negative emotional states are mentioned. These included recognizing that such emotions are a normal and predictable aspect of the parenting experience, particularly in the context of ASD, as well as employing self-reflection practices and adopting alternative perspectives. Furthermore, some parents engaged in a constant cognitive process aimed at making sense of their children's condition; this *deliberate rumination* seemed to function as a strategy that promoted the development of a sense of control over their experience among parents. Discussing openly and disclosing the ASD diagnosis was also an adaptive strategy. This openness was perceived as a key factor in coping with critical moments, essential for promoting understanding by others and facilitating access to emotional and practical support.

A father stated: "*When I meet people, the first thing I say if I am with my son is, "My son is autistic," so that anyone who knows the word autism understands immediately*...".

Finally, many mothers recognized the importance of engaging in self-care activities, such as exercise, social interactions, or simply spending time alone: these moments of detachment from caregiving role were perceived as essential for maintaining a sense of personal identity and autonomy beyond the parental role.

*Self-efficacy.* Many parents expressed a sense of confidence, mastery, and personal satisfaction, often resulting from observing their child's progress and recognizing their own growing competence in supporting that progress. Over time, they reported becoming more skilled and confident in managing ASD and developing a growing sense of trust in their abilities and the strategies they used.

A mother noted: "*I can say that during my first year I attended a "life training course" that was extremely helpful and useful. I had the necessary tools and, gradually, the more I learned, the more confident I felt about what I had to do*".

## Challenges in the adaptation process

Two main themes were identified under this domain: 1) *contextual barriers to adaptation*, and 2) *personal barriers to adaptation*.

**Theme 1: Contextual barriers to adaptation.** This theme consists of three sub-themes: *problems with care and educational services*, *family issues*, and *non-autism-friendly environment*.

*Problems with care and educational services.* Many parents referred significant difficulties with support services. One commonly cited concern was delays and barriers to obtaining a timely and accurate diagnosis. Parents reported that initial preoccupations were sometimes not addressed promptly by paediatricians and healthcare professionals, which resulted in missed opportunities for early intervention. Some participants described the diagnostic process as emotionally traumatic, with some parents feeling overwhelmed when given a diagnosis abruptly and without support.

A mother said: "*I had to fight against windmills, because* [healthcare professionals] *claimed that I was seeing things that didn't exist. And when we finally received the diagnosis, which came late, it was the most difficult moment to accept*".

Other identified problems include extended waiting periods for interventions, inconsistencies within the public health-care system, insufficient communication and coordination among providers, and limited assistance and support for families. As a result, families often resorted to private services at their own expense to obtain appropriate and timely support. Moreover, in relation to the Italian school system, parents highlighted factors such as limited teachers training regarding students with ASD, inadequate communication and coordination among schools, families, and external services, as well as frequent changes in teachers assigned to students with special needs.

*Family issues.* Some parents revealed conflicts with their partner regarding the care and upbringing of their children. These disagreements often stemmed from differences in parenting styles, division of roles, and managing daily responsibilities.

A father said about the mother of his child: *'She is convinced that she does everything for her children. But she is never there, that is, she never participates. She doesn't stay at home with the children, she is struggling... obviously, the divorce has amplified this. In reality, let's say, the divorce has amplified the management problems".*

In some instances, parents experienced denial or avoidance from the other parent regarding their child's diagnosis. Mothers often reported needing to inform fathers about their child's developmental difficulties, even after a formal diagnosis. This reluctance was sometimes linked to minimizing the child's challenges, which in turn could create additional complexities. Other reported a lack of support from families of origin. Furthermore, certain characteristics associated with children with ASD influenced family dynamics and social interactions: difficulties such as delayed language development, rigid routines, frequent emotional outbursts, episodes of aggression, sleep disturbances, and low tolerance to unexpected changes represented challenges for parents.

*Non-autism-friendly environment.* A critical factor in parental adaptation was limited understanding of ASD in the broader social context: in some cases, parents reported that their community lacks adequate knowledge regarding ASD and its complexity, making appropriate recognition and response more challenging. This gap in understanding contributed to negative attitudes, social stigma, and inadequate support systems, thereby heightening the challenges experienced by families.

A father noted: "*This is part of the chapter on -preparing the environment-... this chapter also deals with educating other people. Informing others about autism also means raising awareness; therefore, information and awareness go hand in hand. Because it is easier to be afraid of something you don't know".*

Some parents also mentioned a lack of adequate social support specifically highlighting challenges related to limited availability of practical and financial assistance by local authorities. Reported gaps included services such as transportation, home care, and access to specialized services. Additionally, participants highlighted issues regarding social exclusion and isolation experienced by their children; as children age, parents observed increasing disparities between their child and peers. Consequently, families encountered difficulties accessing inclusive environments, such as birthday parties, playgrounds, or summer camps.

**Theme 2: Personal barriers to adaptation.** This theme consists of two sub-themes: *negative emotional experiences* and *maladaptive coping strategies*. The code *low self-efficacy in parenting* was considered an independent and autonomous sub-theme.

*Negative emotional experiences.* A potentially critical factor was the sense of shame, stigma, and social discomfort that some parents report experiencing. In these cases, parents revealed reluctance to explain their child's atypical behaviours in public, which is often accompanied by feelings of embarrassment.

A mother said: *"It was difficult at first. Sometimes S. would have one of his crises, throwing himself on the ground, I don't know... in a public park where there were people, and I felt... I saw people looking at me... I felt I had to explain to others what I was going through, when I didn't want to explain anything to anyone, I just wanted to try to understand my son… which was devastating".*

Additionally, some parents reported concerns about possible negative or even aggressive reactions from others. Reports from mothers sometimes included accounts for physical and emotional exhaustion, occasionally attributed to insufficient support systems. Finally, parents often expressed a need for respite, indicating a desire for periods of relief from their caregiving responsibilities.

*Maladaptive coping strategies.* Some parents tended to adopt avoidance-oriented coping strategies, such as distraction and denial, when confronted with the challenges of raising a child with ASD. This approach was particularly pronounced among some fathers, who exhibited this coping style by limiting their direct involvement in childcare. In fact, in some cases, parents reported transferring caregiving, educational, and emotional support responsibilities to external agents, including professionals, educational institutions, or extended family members.

For example, a father revealed: "*To improve the situation, you shouldn't think about anything, you shouldn't think about the problem: if you think about the problem, it's over. That's how I see it, you must let it go. You just must do your duty towards that child and not think about the problem; you have to pretend that nothing happened*".

Another critical factor was the rigidity and difficulty in adapting strategies to meet the evolving needs of the child. In these cases, parents described strict adherence to routines, rules, and behavioural expectations across various contexts, even when these strategies proved ineffective. Occasionally, this lack of adaptability resulted in the implementation of overly harsh or punitive disciplinary practices, such as physical punishment. Additionally, some parents experienced negative rumination, characterised by persistent, repetitive thoughts focused on difficulties.

*Low self-efficacy in parenting.* Some parents reported a sense of inadequacy in managing the complexities of caregiving. They described challenges with effective communication, understanding their child's needs, and responding appropriately to behavioural difficulties and crises. These perceptions of ineffectiveness were frequently associated with feelings of frustration and uncertainty, particularly when attempting to identify appropriate strategies, engage the child in meaningful activities, or establish stable and functional daily routines.

One mother said: *"For several years now, I've almost given up on taking him out on my own. Because I'm afraid of handling him... I mean, I can't deal with him, because when he bites me... he bites me and screams, I can struggle all I want, but he keeps going".*

## Discussion

This qualitative study provides an in-depth exploration of the adaptation process among parents of children with ASD, highlighting the complex interplay between emotional distress, personal growth, contextual resources, and barriers. The findings contribute to a growing body of literature that seeks to move beyond a deficit-focused perspective, emphasizing instead the dynamic and multidimensional nature of parental adaptation.

Consistent with previous research [2,3], our results confirm that the period following an ASD diagnosis is characterized by significant emotional distress for parents. Feelings of shock, uncertainty, and anxiety about the future were commonly reported, often accompanied by a sense of resignation or frustration. These experiences underscore the enduring psychological impact of raising a child with ASD and the need for ongoing emotional support for families, rather than viewing adaptation as a finite or linear process.

Importantly, our findings also reveal that many parents experience substantial personal growth and transformation over time. As parents move from initial crisis to acceptance, they often report increased emotional stability, a redefinition of personal values, and a greater sense of empowerment. These results are in line with the broader conceptual framework of post-crisis growth [18,19], which suggests that facing highly challenging experiences may foster positive psychological change and personal development. Parents described developing deeper compassion, enhanced authenticity in relationships, and a renewed sense of purpose, outcomes that are often overlooked in research focused solely on stress and burden.

Additionally, the study highlights that adaptation is not limited to individual psychological change but extends to the family system as a whole. Many parents observed significant improvements in their children's autonomy, communication, and social participation, particularly following targeted interventions. These positive changes contributed to a sense of family empowerment and improved quality of life, reinforcing the reciprocal relationship between child progress and parental adjustment [1,7,24]. This dynamic is consistent with evidence indicating that increased parental knowledge and familiarity with ASD are associated with reduced parental stress [3,27]. Notably, empowerment within families extended beyond the acquisition of practical management skills. Parents described a renewed sense of autonomy and hope, recognising their children's progress as both attainable and meaningful. These findings highlight the pivotal role of learning and mastery in transforming the parenting experience, shifting it from a predominantly stressful endeavour to one characterized by personal growth and satisfaction.

A key contribution of this study is the comprehensive identification of both contextual and personal resources that facilitate adaptation. Social support, encompassing both formal sources (such as professional services and educational settings) and informal networks (including other parents of children with ASD, family, and friends), emerged as a critical protective factor, echoing and extending previous findings [11,22].

Social support played a pivotal role in reducing stress and fostering adaptive coping strategies [3,7,11,32]. Both formal and informal relationships were highlighted as essential, with particular emphasis on the value of friendships, community connections, and peer support from other parents of children with ASD. Such peer relationships were especially significant, as sharing experiences reduced feelings of isolation and provided practical, experience-based knowledge that complemented professional guidance [33]. Additionally, supportive partnerships and the presence of neurotypical siblings contributed to maintaining social engagement and mitigating the risk of excessive focus on ASD within the family. Educational environments and targeted interventions were also recognized as fundamental, with access to qualified professionals and parent training programs enhancing parents' sense of security, competence, and empowerment [24,34–36]. Notably, parents did not perceive these supports as isolated; rather, they described a broader support ecosystem in which interconnected elements, such as social relationships, access to information, involvement in interventions, and inclusive environments, collectively shaped their adaptation experiences.

With respect to parental attributes involved in the adaptation process, the study confirms the fundamental contribution of three interrelated variables in supporting adaptation: (1) *agency*, reflected in proactive strategies and deliberate actions to support the child's development [7,11]; (2) *emotional and reflective management*, involving the regulation of negative emotions, self-reflection, and seeking support within personal and social networks [11,19]; and (3) a *growing sense of confidence and mastery*, or *self-efficacy*, which develops as parents gain experience and witness their children's progress [3,23]. Taken together, these interconnected factors, proactive action, emotional and reflective regulation, and self-efficacy, constitute what can be described as an adaptive and transformative competence. This competence is both adaptive, enabling parents to respond effectively to ongoing challenges, and transformative, fostering personal growth and a redefinition of parental identity. Importantly, this set of resources and skills is dynamic and continually evolving, shaped by both pre-existing personal traits and new strategies acquired through lived experience.

These findings confirm the importance of interventions and policies that not only address child outcomes but also actively promote parental empowerment, skill development, and the cultivation of robust support networks [7]. By

 

recognizing and strengthening this ecosystem of resources, practitioners and policymakers can more effectively support families in navigating the complexities of ASD and fostering resilience and well-being.

Despite the presence of numerous facilitating factors, parents of children with ASD encountered a complex array of barriers that significantly hindered their adaptation process. These obstacles can be conceptualized not as isolated challenges, but as components of a broader barrier ecosystem that interacts with and sometimes counteracts the support systems available to families.

Systemic challenges were frequently reported, including delays in obtaining an ASD diagnosis, fragmented healthcare and educational services, and a general lack of autism-friendly environments. Parents described persistent difficulties in accessing timely and coordinated interventions, often exacerbated by insufficient teacher training, poor communication among service providers, and frequent staff turnover in educational settings [1,3,6,21,24,37]. These systemic shortcomings contributed to ongoing uncertainty and instability for families. Beyond institutional factors, sociocultural barriers such as social stigma and limited public understanding of ASD further intensified parents' feelings of isolation and stress [1,20]. A lack of community awareness often undermined opportunities for families to build supportive networks, constraining their adaptive capacity and reinforcing a sense of exclusion from broader society.

Within the family environment, additional hindering factors emerged. Conflicts between partners, denial or minimization of the diagnosis by one parent, and limited support from extended family members increased the emotional and practical burden on primary caregivers [38]. Even in the presence of supportive resources, these internal family dynamics could impede effective coping and adaptation.

On an individual level, the study identified several personal vulnerabilities that complicated the adaptation process. Negative emotional experiences, such as shame, stigma, and emotional exhaustion, were found to undermine parents' confidence and sense of agency, making it more difficult to proactively address daily challenges [3,8,39]. Maladaptive coping strategies, including avoidance, inflexible behaviour patterns, and persistent negative rumination, further interfered with emotional regulation and problem-solving. Low parental self-efficacy was also associated with increased feelings of inadequacy and uncertainty, potentially limiting parents' perceived ability to positively influence their children's progress [3,11,22].

Taken together, these findings highlight that raising a child with ASD remains a demanding and multifaceted experience, even when personal and environmental resources are present. The interplay between supportive and hindering factors underscores the need for systemic, multi-level interventions that address both structural and interpersonal obstacles. Targeted support should not only aim to dismantle external barriers, such as improving service coordination and reducing stigma, but also address internal vulnerabilities by fostering parental self-efficacy and adaptive coping. By conceptualizing barriers as part of a dynamic ecosystem, rather than isolated obstacles, this study emphasizes the importance of holistic, context-sensitive approaches to supporting families of children with ASD. Interventions and policies must be designed to simultaneously strengthen support systems and mitigate the impact of persistent barriers, ultimately promoting resilience and well-being for both parents and children.

While this study offers valuable insights, several limitations should be acknowledged. Although the primary goal of qualitative research is not to produce generalisable results, the sample was limited to parents of children with moderate to severe ASD in Northern Italy, which may limit the broader applicability of the findings. Furthermore, the cross-sectional design precludes conclusions about changes over time. Future research should include more diverse samples, longitudinal designs, and expanding the focus to incorporate the perspectives of siblings, extended family members, families with more than one child diagnosed with ASD, as well as families in which one or both parents themselves have an ASD diagnosis.

## Conclusions

This study provides a nuanced understanding of the adaptation process among parents of children with ASD, highlighting both the challenges and resources that shape their experiences. The findings suggest that adaptation is a dynamic,

multidimensional process influenced by a complex interplay of personal, familial, and systemic factors. Importantly, the results underscore that adaptation extends beyond individual psychological adjustment to encompass family empowerment, personal growth, and improved quality of life. While parents face significant contextual barriers, including systemic delays, fragmented services, an social stigma, they also draw upon an *ecosystem of supports*, such as meaningful relationships, professional interventions, and social resources. In addition, on a personal level, a combination of agency, emotional regulation, and self-efficacy, an *adaptive and transformative competence*, emerges as a key resource enabling parents to navigate ongoing challenges and foster resilience. These insights have important implications for practice and policy. Interventions should adopt a holistic, family-centred approach that not only addresses stress reduction but also actively promotes empowerment, skill development, and the cultivation of robust support systems. Policies should prioritize the creation of inclusive, coordinated service environments and the reduction of social stigma to enhance family well-being. By recognizing both the vulnerabilities and strengths of families, practitioners and policymakers can more effectively support parents in their journey, ultimately promoting resilience and positive outcomes for children with ASD and their families.

## Supporting information

**S1 Table. Codebook.**
(DOCX)

## Author contributions

**Conceptualization:** Chiara Fante, Alessandro Musetti.

**Data curation:** Fabio Fontana.

**Formal analysis:** Chiara Fante.

**Methodology:** Chiara Fante.

**Project administration:** Alessandro Musetti.

**Supervision:** Raffaele De Luca Picione, Alessandro Musetti.

**Writing – original draft:** Chiara Fante, Alessandro Musetti.

**Writing – review & editing:** Fabio Fontana, Francesca Capelli, Barbara Dioni, Mattia Pezzi, Cinzia Raffin, Raffaele De Luca Picione, Alessandro Musetti.

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
