## [Decision Letter · Decision Letter 0]

24 Jan 2026

PONE-D-25-62770From challenge to growth: A qualitative study of parental adaptation to Autism Spectrum DisorderPLOS One

Dear Dr. Musetti,

Thank you for submitting your manuscript to PLOS ONE. After careful consideration, we feel that it has merit but does not fully meet PLOS ONE’s publication criteria as it currently stands. Therefore, we invite you to submit a revised version of the manuscript that addresses the points raised during the review process.

We look forward to receiving your revised manuscript.

Kind regards,

Gökhan Töret

Academic Editor

PLOS One

Journal Requirements:

Reviewers' comments:

Reviewer's Responses to Questions

**Comments to the Author**

1. Is the manuscript technically sound, and do the data support the conclusions?

Reviewer #1: Yes

Reviewer #2: Partly

2. Has the statistical analysis been performed appropriately and rigorously? 

Reviewer #1: N/A

Reviewer #2: N/A

3. Have the authors made all data underlying the findings in their manuscript fully available?

Reviewer #1: Yes

Reviewer #2: Yes

4. Is the manuscript presented in an intelligible fashion and written in standard English?

Reviewer #1: Yes

Reviewer #2: Yes

5. Review Comments to the Author

Reviewer #1: Materials and Methods

Line 125 – Please clarify the rationale underlying inclusion criterion (b). Specifically, why were Level 1 diagnoses excluded, and why was the age range of 5–11 years selected?

Line 131 – Please explain how the authors verified that no other family members had disabilities or chronic health conditions.

Line 139 – As this is a qualitative study, the use of descriptive statistics (e.g., means and standard deviations) does not appear to be relevant and should be reconsidered.

Line 141 – Please clearly state the number of families who participated in the study. It appears that 23 families were included, but this should be explicitly reported.

Table 1. – Please add information regarding the birth order of the diagnosed child (e.g., eldest, middle, youngest).

Line 177-193 (Data analysis) – Given that the analysis resulted in three overarching conceptual domains, the authors may wish to consider whether the thematic analysis is better described using a different TA approach, such as a framework analysis, which can accommodate both inductive insights from the data and deductive attention to the study’s research questions.

Results

Line 200 – There is a typographical error: “ant” should be corrected to “and”

Line 354 – Please remove the redundant word “and” in the phrase “included increased knowledge and of ASD”

Line 491 - Figure 1 should be placed at the beginning of this section to improve readability and flow.

The descriptions of the following sub-themes should each include at least one illustrative participant quotation:

Line 308 – “resources for managing and understanding”

Line 349 – “educational and professional resources”

Line 405 – “self-efficacy”

Line 416 – “problems with care and educational services”

Line 461 – “negative emotional experiences”

Line 485 – “low self-efficacy in parenting”

Line 429 – “family issues”

Discussion

Line 608 – The authors are encouraged to suggest that future studies include families with more than one child diagnosed with ASD, as well as families in which one or both parents are also diagnosed with ASD.

Reviewer #2: Dear Author,

While the introduction and discussion sections of your study are strong, there are several shortcomings in the methodology and results sections that are considered to require completion.

In the methodology section, it is unclear which qualitative research method was employed. Therefore, it is necessary to clearly specify and describe your research design (e.g., case study, phenomenological research, etc.) in detail. Taking into account the main research need addressed in the introduction, please provide a more detailed explanation of your chosen research model. In addition, it would be more appropriate to provide a justification for your reliability procedures by citing relevant sources. There are numerous approaches to reliability in qualitative research, and the theoretical foundations of the approach you selected should be explained in greater detail. Furthermore, the inter-coder agreement process does not appear to be sufficiently clear. It is recommended that you describe this process in a more explicit and comprehensible manner. In this context, information such as the number of coders, the duration of the coding process, the length of the coded text, and how many times the coders repeated the coding until consensus was reached should be included.

In the results section, although your themes and sub-themes appear to be coherent as a whole, responses supporting some of the sub-themes are not included in your reporting. For example, it is observed that quotations are provided for only two of the sub-themes under the theme Challenges in the Adaptation Process. To present the structure you examined in depth more clearly, it is recommended that you increase the number of quotations.

Within this framework, it is necessary to revise your research article accordingly. Sincerely...

6. PLOS authors have the option to publish the peer review history of their article (what does this mean? ). If published, this will include your full peer review and any attached files.

**Do you want your identity to be public for this peer review?** For information about this choice, including consent withdrawal, please see our Privacy Policy .

Reviewer #1: **Yes:** Tamar Dvir

Reviewer #2: No

---

## [Author Response · Author response to Decision Letter 1]

18 Feb 2026

Reviewer #1

We thank the reviewer for carefully reading the manuscript and for their constructive comments on the methodology and results sections. We have made the following revisions to address the issues highlighted.

Materials and Methods

• Line 125 – Please clarify the rationale underlying inclusion criterion (b). Specifically, why were Level 1 diagnoses excluded, and why was the age range of 5–11 years selected?

The age range of 5 to 11 years was selected to focus on a relatively homogeneous period of development and avoid including parental experiences related to adolescence or the transition to adulthood, which may involve distinct and additional challenges. Furthermore, the exclusion of Level 1 ASD diagnoses was intended to focus on families with significant support needs, which that a potentially impact on their life experience. We have therefore added the following sentences:

(line 129): “The selected age range ensured a relatively homogeneous developmental period, avoiding challenges specific to adolescence or adulthood, which may reflect different parenting experiences. The exclusion of Level 1 ASD diagnoses aligned with the aim of focusing on parents of children with substantial support needs, whose experiences may differ considerably from those of parents of children requiring lower levels of assistance.”.

• Line 131 – Please explain how the authors verified that no other family members had disabilities or chronic health conditions.

We added this clarification on this point:

(line 137): “Compliance with the inclusion criteria was verified through preliminary screening questions before the interview”.

• Line 139 – As this is a qualitative study, the use of descriptive statistics (e.g., means and standard deviations) does not appear to be relevant and should be reconsidered.

We thank the reviewer for this comment. While acknowledging that this is a qualitative study, we consider that maintaining a brief descriptive table helps to contextualise the characteristics of the sample and increases transparency for the reader. In line with the reviewer’s suggestion, we have revised the table by removing variables that were not strictly necessary for this purpose, retaining only those that clarify the key characteristics of the sample. The table is for descriptive purposes only and is not used for inferential purposes.

• Line 141 – Please clearly state the number of families who participated in the study. It appears that 23 families were included, but this should be explicitly reported.

We clarified the point in the sample description.

• Table 1. – Please add information regarding the birth order of the diagnosed child (e.g., eldest, middle, youngest).

We added the information in the Table 1.

• Line 184 (Data analysis) – Given that the analysis resulted in three overarching conceptual domains, the authors may wish to consider whether the thematic analysis is better described using a different TA approach, such as a framework analysis, which can accommodate both inductive insights from the data and deductive attention to the study’s research questions.

We thank the Reviewer for this insightful suggestion. Following this comment, we have revised the Data Analysis section to more accurately describe our method as a Codebook approach to Thematic Analysis [Braun & Clarke, 2021, 2023], which better accommodates the combination of inductive insights and deductive research questions within our three conceptual domains. We revised the section as follow:

(line 182): “This study adopted a Codebook approach to Thematic Analysis (TA), which is part of a broader “family of methods” that share procedural similarities but differ in their underlying research values. This approach was chosen for pragmatic reasons, specifically to facilitate collaborative teamwork and ensure systematic data mapping to achieve the descriptive objectives of the study [29]. Unlike approaches based on coding reliability, the codebook was not used to measure coding accuracy or reliability but served as a flexible tool for tracking and mapping the ongoing analysis [29; 30].

In line with the qualitative values of the ‘Big Q’ proposed by Braun & Clarke [29; 30], the researcher’s subjectivity was considered a fundamental resource for interpretation rather than a bias to be contained. Consequently, formal indices of agreement between coders or consensual coding were not calculated, as they were not aligned with the qualitative research values guiding this study; instead, the research team used reflective dialogue to obtain a multifaceted understanding of the data.

During the early familiarisation phase, researchers read the interview transcripts multiple times. Four interviews were therefore selected as particularly suitable for preliminary coding, given their narrative richness (two with mothers and two with fathers). Each member of the research team independently coded these transcripts, generating preliminary conceptual labels (codes) that captured key elements of meaning. Through reflexive dialogue and peer debriefing, these initial labels were developed into a preliminary codebook.

The full dataset was then coded by two authors (C.F and F.F.) using this evolving codebook, which was iteratively refined to chart and map the developing analysis [29]. Coding decisions were regularly discussed and reviewed in team meetings and peer debriefing sessions, adopting a collaborative and reflexive approach to data interpretation [30].

At the conclusion of the coding process, similar or related codes were grouped into preliminary sub-themes and themes, which were developed inductively from the coded data and then collaboratively reviewed and refined. Specifically, the themes were actively generated and constructed by the research team, reflecting the researchers’ active role in the analytic process. In line with the descriptive focus of the study, these themes were conceptualized as “topic summaries”, structured overviews that unite participants accounts around specific conceptual domains. These summaries were organized into three broad domains- Outcomes, Resources, and Challenges- to provide a clear and transparent narrative of the adaptation process.

The final coding system comprised 67 codes, grouped into 7 main themes. Data analysis was conducted using MAXQDA, a software application for qualitative data analysis [30]. The complete system of codes, including definitions and organisation into sub-themes and themes, is available in the supplementary materials (S1 Table).

Results

• Line 200 – There is a typographical error: “ant” should be corrected to “and”

Thank you for pointing this out. We have corrected the error.

• Line 354 – Please remove the redundant word “and” in the phrase “included increased knowledge and of ASD”

Removed.

• Line 491 - Figure 1 should be placed at the beginning of this section to improve readability and flow.

Thank you for the useful suggestion. We have moved the figure to the beginning of the section and modified the opening sentence as follows: (line 218): “The themes and sub-themes identified and organized into the three broad conceptual domains are represented visually in Figure 1 and described in detail below”.

• The descriptions of the following sub-themes should each include at least one illustrative participant quotation

We added the following quotations relating to the requested sub- themes:

– “resources for managing and understanding”

(line 332): One father said: “Augmentative communication with images has had a huge impact. We noticed that S. is very visual, so showing him activities through images or written words has been a big help.”

– “educational and professional resources”

(line 374): A mother said: “Relying on experts is a great help, especially in difficult times, when you feel unable to deal with a critical situation involving your child: knowing the strategies suggested by professionals and knowing that you are able to manage the situation...”

– “self-efficacy”

(line 463): A mother noted: “I can say that during my first year I attended a “life training course” that was extremely helpful and useful. I had the necessary tools and, gradually, the more I learned, the more confident I felt about what I had to do.”

– “problems with care and educational services”

(line 450): A mother said: “I had to fight against windmills, because [healthcare professionals] claimed that I was seeing things that didn't exist. And when we finally received the diagnosis, which came late, it was the most difficult moment to accept”.

– “negative emotional experiences”

(line 500): A mother said: “It was difficult at first. Sometimes S. would have one of his crises, throwing himself on the ground, I don't know... in a public park where there were people, and I felt... I saw people looking at me... I felt I had to explain to others what I was going through, when I didn't want to explain anything to anyone, I just wanted to try to understand my son… which was devastating.”

– “low self-efficacy in parenting”

(line 531): One mother said: “For several years now, I’ve almost given up on taking him out on my own. Because I’m afraid of handling him... I mean, I can’t deal with him, because when he bites me... he bites me and screams, I can struggle all I want, but he keeps going.”

– “family issues”

(line 463): A father said about the mother of his child: ‘She is convinced that she does everything for her children. But she is never there, that is, she never participates. She doesn't stay at home with the children, she is struggling... obviously, the divorce has amplified this. In reality, let's say, the divorce has amplified the management problems.”

Discussion

• Line 608 – The authors are encouraged to suggest that future studies include families with more than one child diagnosed with ASD, as well as families in which one or both parents are also diagnosed with ASD.

We have added this part of the sentence (underlined; line 648): “Future research should include more diverse samples, longitudinal designs, and expanding the focus to incorporate the perspectives of siblings, extended family members, families with more than one child diagnosed with ASD, as well as families in which one or both parents themselves have an ASD diagnosis.”

Reviewer 2

• While the introduction and discussion sections of your study are strong, there are several shortcomings in the methodology and results sections that are considered to require completion.

We would like to thank the Reviewer for all valuable comments and suggestions, which will help us in improving the quality of the manuscript. We have made the following revisions to address the issues highlighted.

Following the suggestion, we have specified that this study adopts a qualitative descriptive research design using a Codebook approach to Thematic Analysis. We have expanded the methodology section to better explain this model aligns with the exploratory nature of our research questions.

Regarding the request for justification of our reliability procedures, we have added a detailed explanation citing Braun & Clarke (2021, 2023) to clarify that our study is situated within a Big Q qualitative framework and justify our decision to prioritize methodological integrity and reflexive dialogue.

• In the methodology section, it is unclear which qualitative research method was employed. Therefore, it is necessary to clearly specify and describe your research design (e.g., case study, phenomenological research, etc.) in detail. Taking into account the main research need addressed in the introduction, please provide a more detailed explanation of your chosen research model. In addition, it would be more appropriate to provide a justification for your reliability procedures by citing relevant sources. There are numerous approaches to reliability in qualitative research, and the theoretical foundations of the approach you selected should be explained in greater detail. Furthermore, the inter-coder agreement process does not appear to be sufficiently clear. It is recommended that you describe this process in a more explicit and comprehensible manner. In this context, information such as the number of coders, the duration of the coding process, the length of the coded text, and how many times the coders repeated the coding until consensus was reached should be included.

We thank the reviewer for highlighting the need to clarify the qualitative research design and analytical procedures adopted in the study. Following the suggestion, we specified that this study adopts a qualitative descriptive research design using a Codebook approach to thematic analysis. We expanded the methodology section to better explain how this choice is in line with the exploratory nature of our research questions. Regarding the request for justification of our reliability procedures, we added a detailed explanation citing Braun & Clarke (2021, 2023) to clarify that our study falls within a Big Q qualitative framework and to justify our decision to prioritise methodological integrity and reflective dialogue. We have also provided a more detailed description of the collaborative coding process. We have included information about the coders, the iterative nature of coding code development, and the collaborative reflection process used to achieve a thorough understanding of the data.

We revised the section as follow:

(line 182): “This study adopted a Codebook approach to Thematic Analysis (TA), which is part of a broader “family of methods” that share procedural similarities but differ in their underlying research values. This approach was chosen for pragmatic reasons, specifically to facilitate collaborative teamwork and ensure systematic data mapping to achieve the descriptive objectives of the study [29]. Unlike approaches based on coding reliability, the codebook was not used to measure coding accuracy or reliability but served as a flexible tool for tracking and mapping the ongoing analysis [29; 30].

In line with the qualitative values of the ‘Big Q’ proposed by Braun & Clarke [29; 30], the researcher’s subjectivity was considered a fundamental resource for interpretation rather than a bias to be contained. Consequently, formal indices of agreement between coders or consensual coding were not calculated, as they were not aligned with the qualitative research values guiding this study; instead, the research team used reflective dialogue to obtain a multifaceted understanding of the data.

During the early familiarisation phase, researchers read the interview transcripts multiple times. Four interviews were therefore selected as particularly suitable for preliminary coding, given their narrative richness (two with mothers and two with fathers). Each member of the research team independently coded these transcripts, generating preliminary conceptual labels (codes) that captured key elements of meaning. Through reflexive dialogue and peer debriefing, these initial labels were developed into a preliminary codebook.

The full dataset was then coded by two authors (C.F and F.F.) using this evolving codebook, which was iteratively refined to chart and map the developing analysis [29]. Coding decisions were regularly discussed and reviewed in team meetings and peer debriefing sessions, adopting a collaborative and reflexive approach to data interpretation [30].

At the conclusion of the coding process, similar or related codes were grouped into preliminary sub-themes and themes, which were developed inductively from the coded data and then collaboratively reviewed and refined. Specifically, the themes were actively generated and constructed by the research team, reflecting the researchers’ active role in the analytic process. In line with the descriptive focus of the study, these themes were conceptualized as “topic summaries”, structured overviews that unite participants accounts around specific conceptual domains. These summaries were organized into three broad domains- Outcomes, Resources, and Challenges- to provide a clear and transparent narrative of the adaptation process.

The final coding system comprised 67 codes, grouped into 7 main themes. Data analysis was conducted using MAXQDA, a

---

## [Editor Report · Decision Letter 1]

24 Feb 2026

PONE-D-25-62770R1From challenge to growth: A qualitative study of parental adaptation to Autism Spectrum DisorderPLOS One

Dear Dr. Musetti,

Thank you for submitting your manuscript to PLOS ONE. After careful consideration, we feel that it has merit but does not fully meet PLOS ONE’s publication criteria as it currently stands. Therefore, we invite you to submit a revised version of the manuscript that addresses the points raised during the review process.

The reviewers’ comments have been carefully addressed, and the manuscript has improved substantially. The study is promising and suitable for publication pending a few minor clarifications.

Please address the following points:

Please briefly summarize the analytic approach (codebook thematic analysis and big-Q framework) to ensure clarity for readers with diverse methodological backgrounds.

Please ensure that the data availability statement clearly specifies the minimal dataset available and the conditions under which qualified researchers may request access to additional de-identified data, in line with PLOS ONE requirements.

We look forward to receiving your revised manuscript.

Kind regards,

Gökhan Töret

Academic Editor

PLOS One
---

## [Author Response · Author response to Decision Letter 2]

24 Feb 2026

Response to the Editor

Dear Editor,

We sincerely thank you for your constructive comments and the opportunity to revise our manuscript. We have carefully addressed each point, as detailed below.

Editor Comment 1:

“Please briefly summarize the analytic approach (codebook thematic analysis and Big-Q framework) to ensure clarity for readers with diverse methodological backgrounds.”

Response:

We thank the Editor for this helpful suggestion. We have revised the manuscript to strengthen the methodological description by clearly outlining how the codebook approach to Thematic Analysis and the Big Q qualitative framework guided our analytic process. Specifically, we expanded the Data Analysis section (now found in lines 182–196).

Editor Comment 2:

“Please ensure that the data availability statement clearly specifies the minimal dataset available and the conditions under which qualified researchers may request access to additional de-identified data, in line with PLOS ONE requirements.”

Response:

We fully revised the Data Availability Statement to comply with PLOS ONE’s policy for studies involving sensitive qualitative human participant data.

Once again, we thank you for your thoughtful guidance, which has strengthened the clarity and transparency of our manuscript. We hope the revisions meet your expectations and remain available for any additional clarification.

Kind regards,

Alessandro Musetti

---

## [Editor Report · Decision Letter 2]

2 Mar 2026

From challenge to growth: A qualitative study of parental adaptation to Autism Spectrum Disorder

PONE-D-25-62770R2

Dear Dr. Musetti,

I appreciate the attention you have given to the revision. I have reviewed the updated manuscript and your detailed responses to the editorial comments. I am pleased to confirm that the requested revisions have been addressed.

We’re pleased to inform you that your manuscript has been judged scientifically suitable for publication and will be formally accepted for publication once it meets all outstanding technical requirements.

Kind regards,

Gökhan Töret

Academic Editor

PLOS One
---

## [Editor Report · Acceptance letter]

PONE-D-25-62770R2

PLOS One

Dear Dr. Musetti,

I'm pleased to inform you that your manuscript has been deemed suitable for publication in PLOS One. Congratulations! Your manuscript is now being handed over to our production team.

Kind regards,

on behalf of

Dr. Gökhan Töret

Academic Editor

PLOS One